# Antibacterial and Antiviral Activities of Local Thai Green Macroalgae Crude Extracts in Pacific White Shrimp (*Litopenaeus vannamei*)

**DOI:** 10.3390/md18030140

**Published:** 2020-02-27

**Authors:** Nawanith Klongklaew, Jantana Praiboon, Montakarn Tamtin, Prapansak Srisapoome

**Affiliations:** 1Laboratory of Aquatic Animal Health Management, Department of Aquaculture, Faculty of Fisheries, Kasetsart University, Bangkok 10900, Thailand; nawanith.k@ku.th; 2Department of Fishery Biology, Faculty of Fisheries, Kasetsart University, Bangkok 10900, Thailand; ffisjtn@ku.ac.th; 3Phetchaburi Coastal Aquaculture Research and Development Center, Department of Fisheries, Ministry of Agriculture and Cooperatives, Phetchaburi 76100, Thailand; mtamtin@hotmail.com

**Keywords:** antibacterial activity, antiviral activity, white spot syndrome virus, WSSV, yellow head virus, YHV, green algae, Pacific white shrimp, *Litopenaeus vannamei*

## Abstract

Macroalgae are potentially excellent sources of bioactive secondary metabolites useful for the development of new functional ingredients. This study was conducted to determine the antimicrobial efficacy of the hot water crude extracts (HWCEs) of three species of local Thai green macroalgae *Ulva intestinalis* (Ui), *U. rigida* (Ur), and *Caulopa lentillifera* (Cl) and a commercial ulvan from *U. armoricana* (Ua). Chemical analysis indicated that the HWCE of Ur showed the highest sulfate content (13.9% ± 0.4%), while that of Ua contained the highest uronic acid and carbohydrate contents (41.47% ± 4.98% and 64.03% ± 2.75%, respectively), which were higher than those of Ur (32.75% ± 1.53% and 51.02% ± 3.72%). Structural analysis of these extracts by Fourier-transform infrared (FTIR) spectroscopy revealed that these HWCEs are complex with a signal at 1250 cm^−1^ corresponding to S=O stretching vibrations, while the signals at 850 cm^−1^ were attributed to the C–O–S bending vibration of the sulfate ester in the axial position. These HWCEs showed the growth suppression against some pathogenic *Vibrio* spp. Interestingly, the HWCEs from Ui at concentrations of 5 and 10 mg/mL completely inhibited white spot syndrome virus (WSSV) in shrimp injected with HWCE–WSSV preincubated solutions. This inhibitory effect was further confirmed by the reduction in viral loads and histopathology of surviving and moribund shrimp.

## 1. Introduction

The Pacific white shrimp (*Litopenaeus vannamei*) is a dominant crustacean in coastal aquaculture around the world and an important source of export earnings for developing countries in Asia and Latin America. The worldwide marine shrimp aquaculture industry experienced rapid changes over the past few decades. While the global production of captured shrimp was fairly stable in recent years, that of cultured Pacific white shrimp expanded rapidly, from approximately 3,238,000 tons in 2012 to approximately 4,156,000 tons in 2016. Recently, disease outbreaks and climate change are an ongoing challenge for some large Asian shrimp aquaculture producers, particularly in Thailand and China [1].

The most important causative agents of shrimp diseases are currently pathogenic bacteria and viruses [2]. White spot syndrome virus (WSSV), which belongs to the family Nimaviridae, genus Whispovirus, has a large circular double-stranded DNA genome with different sizes (292.9–307.2 kb), large virions (80–120 × 250–380 nm) that are rod- to elliptical-shaped, and a trilaminar envelope [2]. The signs observed in WSSV-infected shrimp are white deposits of calcium 0.5–2.0 mm in diameter on the cuticle of the shrimp cephalothorax or carapace [3], and infected shrimp display reddish to pinkish discoloration, with presence of circular white patches of 0.5–3.0 mm in diameter most prominent in the cuticle of cephalothorax and abdominal segments; they also become lethargic, cease feeding, and normally congregate at the pond edges [4]. WSSV was determined to be the cause of mass mortalities (often reaching 80%–100% of affected populations) of cultured shrimp in Asia and eventually the Americas [5]. Another serious shrimp disease in Asia is caused by yellow head virus (YHV). YHV is an enveloped, rod-shaped, single-stranded RNA (ssRNA) virus in the family Roniviridae and the order Nidovirales [2]. The virions arise from longer, filamentous nucleocapsids (approximately 15 nm × 130–800 nm). Yellow head disease (YHD) is characterized by high and rapid mortality that is sometimes accompanied by yellowing of the cephalothorax (for which the disease was named) and general bleaching of the body [2]. Subsequent mass mortality of shrimp occurs within 2–3 days of the first appearance of signs of the disease [6]. Furthermore, other main causes of diseases in shrimp culture systems are attributed to vibriosis, that recognize as etiological agents 11 species of the pathogenic *Vibrio* spp. such as *Vibrio harveyi* and *V. alginolyticus*, as well as acute hepatopancreatic necrosis disease (AHPND) and early mortality syndrome (EMS) caused by *V. parahaemolyticus* [7]. Mortalities caused by AHPND generally occur within the first five weeks of stocking a pond with postlarvae, although there is a report of AHPND killing late-stage juveniles cultured for ~3 months, and clinical signs of the disease include lethargy, slow growth, an empty stomach and midgut, and a pale to white atrophied hepatopancreas, with dead shrimp often amassing at the pond bottom [8].

Antibiotics and chemotherapeutics were used to control microbial infections in aquaculture for more than 20 years [9]. However, the use of antibiotics is not recommended in shrimp farming because the resulting residues endanger consumers, and the overuse and misuse of antibiotics increases the threat of antibiotic-resistant bacteria [10]. Hence, shrimp health management by natural substances is used as a necessary strategy for the development of the shrimp culture industry.

In recent decades, the discovery of metabolites from marine resources showing biological activity increased significantly. Among these marine resources, algae are a valuable source of structurally diverse bioactive compounds that can possess both immunostimulatory and antimicrobial effects [11]. Marine algae, in particular, are a rich source of functional metabolites such as polysaccharides, proteins, peptides, lipids, amino acids, polyphenols, vitamins, and minerals [12]. Basically, algae are classified into three major groups based on their pigmentation, types of storage material, and the composition of their cell-wall polysaccharides: (1) brown algae (*Phaeophyceae*), (2) red algae (*Rhodophyceae*), and (3) green algae (*Chlorophyceae*) [12,13,14].

Within marine macroalgae metabolites, polysaccharides are polymers of simple sugars (monosaccharides) linked together by glycosidic bonds [15]. Marine algae polysaccharides (MAPs) are the most important type of biological molecules contained in large amounts of marine extracts [16], and MAPs with high sulfate functionalization are generally known as sulfate polysaccharides. The type of sulfate polysaccharides in marine algae, such as alginate, fucoidan, agar, carrageenan, porphyran, laminarin, galactan, and ulvan, differs depending on the taxonomic group. These sulfate polysaccharides exhibit many beneficial biological functions and pharmacological activities, such as anticoagulant, antiviral, antioxidant, antitumor, anticancer, anti-inflammatory, anti-hyperlipidemia, antimicrobial, and immunomodulatory activities [12,13,14,15,16,17]. Additionally, in recent years, many countries around the world began applying various MAPs to effectively promote the growth and immune responses of aquatic animals. Some of these are also applied to control various pathogens and reduce the use of antibiotics and chemotherapies in aquaculture farms.

In Thailand, the biomass of marine macroalgae is mainly represented by a few species of Rhodophyta, including *Gracilaria fisheri,* and Chlorophyta, including *Caulopa lentillifera, Ulva intestinalis*, and *U. rigida* [18,19,20,21,22,23,24]. However, there were very few studies of algal applications in aquaculture in Thailand. Most of these algae are only applied for water quality improvement and recycling of nutrients in shrimp farms [20]. Because of the low rate at which they are consumed, there is considerable algal waste each year [23]. Therefore, to properly increase the utilization of these algae, three local Thai macroalgae, including *C. lentillifera, U. intestinalis*, and *U. rigida,* were used for the extraction of potentially bioactive substances, including mainly sulfate polysaccharides. These substances are known to be useful for enhancing immune and antimicrobial properties in several aquatic animals and may provide a new functional strategy for pharmaceutical treatments in Thai shrimp culture. For this reason, the aims of this study were to analyze the chemical compositions of hot water crude extracts (HWCEs) from local Thai macroalgae and investigate their antibacterial and antiviral properties in Pacific white shrimp.

## 2. Results

### 2.1. Chemical Content and Structural Analyses of Crude Polysaccharides from Three Local Thai Green Algae

Sulfate content analysis of the HWCEs from three species of local Thai green macroalgae and a commercial purified ulvan from *U. armoricana* (Ua) showed that the HWCEs of *U. rigida* (Ur) contained a significantly higher sulfate content than those of other species at 13.89% ± 0.38%, followed by 11.50% ± 0.44%, 11.01% ± 0.29%, and 7.29% ± 0.28% in Ua, *U. intestinalis* (Ui), and *C. lentillifera* (Cl), respectively. The uronic acid content showed the maximal value in Ua at 41.47% ± 4.89%, which was significantly higher than that of Ur, Ui, and Cl with values of 32.76% ± 1.53%, 29.58% ± 1.66%, and 9.22% ± 0.56%, respectively (*p* < 0.05). Additionally, Ua contained significantly higher carbohydrate contents of 64.03% ± 2.75% than those of Ur, Ui, and Cl (Table 1).

The structural compositions of the HWCEs of three local species of Thai green algae were analyzed by Fourier-transform infrared spectroscopy (FTIR). A typical infrared spectrum of ulvan showed strong absorbance at approximately 1650, 1250, and 1070 cm^−^^1^ and weak absorbance at approximately 1400, 850, and 790 cm^−^^1^ [25,26]. The first band represented a broad signal at 3434 cm^−^^1^, attributed to the stretching vibrations of the OH group, and a small signal at 2939 cm^−^^1^ was attributed to the stretching vibration of the CH group [27,28]. There were two carboxylate groups (C=O) indicating an asymmetric stretching band of uronic acid near 1635 cm^−^^1^ and a weaker symmetric stretching band at 1425 cm^−^^1^ [25,29]. Sulfate ester (S=O) showed a major band at approximately 1250 cm^−^^1^. Two minor bands at 850 and 790 cm^−^^1^ corresponded to the C–O–S bending vibration of sulfate, and all ulvan spectra presented one strong band at approximately 1052 cm^−^^1^, likely due to C-O stretching from two main sugars, rhamnose and glucuronic acid. The signal between 1200–1000 cm^−^^1^ was dominated by sugar ring and glycosidic bond C–O stretching vibrations (Figure 1). The structure of the HWCEs from the algae had an infrared spectrum similar to that of the commercial purified ulvan from Ua (Figure 1).

### 2.2. Acute Toxicity of the HWCEs from Local Thai Green Macroalgae

Pacific white shrimp free of bacterial and viral infections were used to evaluate the acute toxicity of the HWCEs from three species of green macroalgae and the commercial ulvan from *U. armoricana* by intramuscular injection at 0, 1, 5, and 10 mg/mL, followed by seven days of observation of moribund or dead shrimp. All shrimp from every experimental group survived (100%) with no clinical signs.

### 2.3. Antibacterial Activity of the HWCEs from Local Thai Green Macroalgae 

The antibacterial activity against several shrimp bacterial pathogens including *V. alginolyticus*, *V. vulnificus*, *V. harveyi*, *V. parahaemolyticus*, and *V. parahaemolyticus* (AHPND strain) was determined through the minimal inhibitory concentrations (MICs) of the HWCEs at 0–128 μg/mL and compared with that of enrofloxacin (128 μg/mL) as a positive control (Figure 2). The experiment was completed after 48 h, at which time the absorbance of bacterial growth was plotted. The results revealed that no antibacterial effects were observed on any shrimp pathogenic bacterium for all HWCEs and the commercial ulvan. No MICs displayed positive responses to extracts because the absorbance of each concentration of each HWCE was clearly increased, similar to that of the control. Slight inhibition was found in the Ur experiment, in which all concentrations of HWCEs could suppress *V. vulnificus* growth only during hours 6–18 (Figure 2b). In contrast, at all concentrations of all test substances, slight increases in the growth rates of *V. alginolyticus* and *V. vulnificus* and strong increases in the growth rates of *V. parahaemolyticus* and *V. parahaemolyticus* AHPND strain were observed, as indicated by the significantly higher absorbances compared with the positive and negative controls (Figure 2a–d).

### 2.4. Antiviral Activity of the HWCEs from Local Thai Green Algae

#### 2.4.1. Shrimp Mortality

Anti-WSSV activity of the HWCEs in Pacific white shrimp was evaluated by injection of the shrimp with extract–virus mixtures of 0, 1, 5, and 10 mg/mL of HWCEs from three species of green macroalgae and a commercial ulvan from *U. armoricana* and a lethal dose concentration (10^7.75^ LD_50_/mL at 96 h) of WSSV.

In the Ui experiment, all concentrations of the Ui HWCE resulted in significantly lower cumulative mortality than positive control shrimp groups (*p* < 0.05). The mortality rate in the positive control group (WSSV-injected) rapidly increased to 55.0% ± 35.4% at day four after injection and continued to reach 100.0% ± 0.0% at day 10, while the negative control (phosphate-buffered saline (PBS)-injected) group and Ui HWCE–WSSV-injected groups showed mortality rates of 0.0% ± 0.0%, 5.0% ± 7.1%, 6.1% ± 10.5%, and 10.0% ± 10.0%, respectively (*p* < 0.05) (Figure 3a).

In the Ur experiment, all WSSV-injected groups showed rapid mortality at day three after injection, and the highest mortality rates of during days 6–10, except for the negative control group (Figure 3b). At the end of the challenge test, the positive control and Ur HWCE–WSSV-injected groups exhibited the highest mortalities of 100.0% ± 0.0%, and 100.0% ± 0.0%, 96.7% ± 5.8%, and 100.0% ± 0.0%, respectively, while the negative control group expressed the lowest mortality of 5.0 ± 7.1% (Figure 3b). 

Similar to the Ui experiment, in the Cl experiment the injection of extracts from Cl showed very good results. Compared with the positive control, shrimp injected with WSSV and 1–10 mg/mL of Cl HWCE exhibited significantly lower mortality of 0.0%–6.7%, which was less than that of the positive control, during all experimental periods (Figure 3c). At day 10, the positive control showed significantly higher mortality than the negative control and the Cl HWCE–WSSV-injected groups with mortality rates of 100.0% ± 0.0%, 5.0% ± 7.1%, 0.0% ± 0.0%, 0.0% ± 0.0%, and 6.7% ± 11.5%, respectively (*p* < 0.05).

Interestingly, the preincubated commercial ulvan Ua–WSSV-injected group (Figure 3d) exhibited very strange results since the Ua concentration at 1 and 5 mg/mL showed significantly lower mortality than the positive control throughout the experiment (*p* < 0.05). However, the shrimp injected with preincubated commercial ulvan Ua–WSSV at 10 mg/mL exhibited a high mortality rate that was not significantly different from that of the positive control at almost of the time periods (*p* > 0.05). At the end of a 10-day trial, the shrimp groups injected with 1 and 5 mg/mL Ua–WSSV solutions showed mortality rates of 0.0% ± 0.0% and 0.0% ± 0.0%, respectively, which were significantly lower than those of the positive control shrimp and shrimp injected with Ua–WSSV at 10 mg/mL, which clearly showed mortality rates of 100.0% (*p* < 0.05).

The anti-YHV activity of the HWCEs in Pacific white shrimp was also evaluated by injection of the shrimp with preincubated solutions of 0, 1, 5, and 10 mg/mL of the HWCEs from three species of green macroalgae and commercial ulvan from *U. armoricana* and a lethal dose concentration (10^9.52^ LD_50_/mL at 96 h) of YHV.

The results of the Ui experiment showed that all concentrations of Ui HWCE resulted in significantly lower cumulative mortality rates than the positive control shrimp groups (*p* < 0.05). The mortality rate in the positive control group (YHV-injected) rapidly increased to 95.0% ± 7.1% at day four after injection and reached 100.0% ± 0.0% at day 10, while the negative control (PBS-injected) group and Ui HWCE–YHV-injected groups showed mortality rates of 0.0% ± 0.0%, 9.4% ± 9.1%, 13.3% ± 5.8%, and 26.1% ± 6.8%, respectively (*p* < 0.05) (Figure 4a).

In the Ur experiment, the 1 and 5 mg/mL concentrations of HWCEs showed cumulative mortality rates lower than those of the positive control groups (*p* < 0.05) throughout the experimental period, except for the 10 mg/mL Ur HWCE–YHV-injected group. The mortality rate of the 10 mg/mL Ur HWCE–YHV-injected group suddenly increased to 70.0% ± 14.1% at day six after injection. The positive control and 10 mg/mL Ur HWCE–YHV-injected group exhibited the highest mortality of 100.0% ± 0.0% at the end of the experiment (Figure 4b).

In the Cl experiment, all YHV-injected groups showed a gradual increase in mortality at four days post injection and rapid mortality during days 5–10, except for the negative control group (Figure 4c). At day 10, the positive control and Cl HWCE–YHV-injected groups exhibited mortality rates of 100.0% ± 0.0%, and 6.7% ± 5.8%, 85.0% ± 21.1%, and 90.0% ± 14.1%, respectively, while the negative control group exhibited a mortality rate of 0.0 ± 0.0%. Likewise, the cumulative mortality of the preincubated commercial ulvan Ua–YHV-injected group was higher than that of the 5 mg/mL Ua HWCE–YHV-injected group, which was higher than that of the positive control group on day three after injection. In addition, all of the Ua HWCE–YHV-injected groups presented a rapid mortality rate during days 3–10 (Figure 4d). At day 10, the positive control and the Ua HWCE–YHV-injected groups had significantly higher mortality rates (100.0% ± 0.0%, and 76.7% ± 32.1%, 96.7% ± 5.8%, and 95.0% ± 7.1%, respectively) than the negative control group (*p* < 0.05).

#### 2.4.2. Viral Loading Assay

The viral loads of WSSV were investigated in the Pacific white shrimp by absolute quantitative real-time qPCR in the gills and intestine of six randomized samples collected from newly dead and surviving shrimp in each HWCE experiment (Figure 5). The moribund or newly dead shrimp were collected throughout the experimental period, and the surviving shrimp were collected at the end of the experiment. These samples were randomly chosen for viral loading and histopathological assays.

In the gills of the Ui shrimp, six newly dead shrimp from the positive control and six surviving shrimp injected with different concentrations of Ui HWCE and WSSV were randomly selected. In the dead shrimp from the positive control, 6.28 ± 6.39 log_10_ copy number/µg of total DNA was detected, which was significantly higher than that of all the surviving shrimp from the preincubated 1, 5, and 10 mg/mL Ui HWCE–WSSV groups, in which WSSV DNA was not detected (*p* < 0.05). The levels in the latter groups were similar to those of some dead shrimp found in these groups that may have died from cannibalism (data not shown) (Figure 5a1). Similar results were observed in the Cl experiment. The positive control shrimp exhibited a high viral load of 6.28 ± 6.39 log_10_ copies number/µg of total DNA, which was significantly higher than those of Cl HWCE–WSSV-injected groups at 1, 5, and 10 mg/mL, in which no WSSV DNA was found (*p* < 0.05) (Figure 5c1).

In the Ur experiment, the viral load in the gills of dead shrimp in different groups that were injected with 1–10 mg/mL Ur HWCE–WSSV solutions was found to be 6.28 ± 6.39 log_10_ copies number/µg of total DNA, which was significantly higher than those of the other Ur HWCE-WSSV-injected groups, which showed approximately 2.60 ± 6.39, 1.96 ± 2.12, and 6.59 ± 6.87 log_10_ copies number/µg of total DNA, respectively (*p* < 0.05). Additionally, a surviving shrimp from the 5 mg/mL Ur HWCE–WSSV-injected group also showed a high value of 2.88 ± 0.00 log_10_ copies number/µg of total DNA (Figure 5b1). These results were similar to those of the commercial Ua experiment, which found that the viral load of WSSV in the dead shrimp of the positive control group was significantly higher than that in the other Ua (1–10 mg/mL)-WSSV-injected groups with 1.58 ± 1.46, 2.48 ± 2.58, and 3.01 ± 3.35 log_10_ copies number/µg of total DNA, respectively (*p* < 0.05).

For the WSSV load in the intestines of dead and surviving shrimp, the viral loads from the dead shrimp in the positive control and 1 mg/mL Ui HWCE–WSSV-injected groups (4.39 ± 4.61 and 6.94 ± 7.09 log_10_ copies/µg of total DNA) were significantly higher than those of the surviving shrimp from the preincubated 5 and 10 mg/mL Ui HWCE–WSSV groups, which showed no WSSV DNA (*p* < 0.05) (Figure 5a2).

In the Ur experiment, dead shrimp in the 1 mg/mL Ur HWCE–WSSV-injected group contained a WSSV load of 0.91 ± 1.01 log_10_ copies number/µg of total DNA, which was significantly different from the positive control and groups infect with Ur HWCE–WSSV at 5 and 10 mg/mL, which had viral loads of 4.39 ± 4.61, 1.61 ± 0.69, and 5.23 ± 5.30, respectively (*p* < 0.05) (Figure 5b2). In the Cl experiment, all concentrations of Cl HWCE–WSSV-injected groups expressed significantly lower viral loads (0.00 ± 0.00, 0.00 ± 0.00, and 0.79 ± 0.99, respectively) than the positive control (4.39 ± 4.61 log_10_ copies number/µg of total DNA (*p* < 0.05) (Figure 5c2). This datum was similar to that of the Ua commercial ulvan experiment, in which all Ua HWCE–WSSV-injected groups expressed significantly lower viral loads (0.91 ± 0.93, 0.00 ± 0.00, and 2.02 ± 2.11, respectively) than the positive control (*p* < 0.05) (Figure 5d2).

The YHV load was also investigated in the gills and hepatopancreas of six randomized samples collected from newly dead and surviving Pacific white shrimp. For the Ui experiment, the dead positive control shrimp contained 5.85 ± 0.72 log_10_ copies number/µg of total complementary DNA (cDNA) in their gills, which was significantly higher than that in all the newly dead and surviving shrimp from the preincubated Ui HWCE–YHV groups at 1, 5, and 10 mg/mL, which was 1.95 ± 0.34, 2.12 ± 1.02, and 1.56 ± 1.30, and 2.59 ± 1.20, 2.35 ± 1.23, and 2.04 ± 0.50 log_10_ copies number/µg of total cDNA, respectively (*p* < 0.05) (Figure 6a1). In the Ur experiment, the YHV load in the gills of the newly dead and surviving shrimp from the 1 and 5 mg/mL Ur HWCE–YHV-injected groups and the surviving shrimp of the 10 mg/mL Ur HWCE–YHV-injected group were significantly lower than those of the positive control group. However, the viral load of the newly dead shrimp of the 10 mg/mL Ur HWCE–YHV-injected group was not significantly different from that of the positive control (6.16 ± 0.48 log_10_ copies number/µg of total cDNA (Figure 6b1).

Similarly, in the C1 experiment, the viral loads in gills of newly dead and surviving shrimp from the 1–10 mg/mL Ur HWCE–YHV group were significantly lower than those of the positive control group (*p* < 0.05), except for the newly dead shrimp of the 5 and 10 mg/mL Cl HWCE–YHV-injected groups, which were higher than those of the other Cl HWCE–YHV-injected groups, with viral loads of approximately 6.18 ± 0.20 and 6.01 ± 0.79 log_10_ copies number/µg of total cDNA (Figure 6c1). In particular, the YHV loads in freshly dead shrimp of the 1–10 mg/mL Ua HWCE–YHV-injected groups were 5.39 ± 1.27, 6.13 ± 0.44, and 5.15 ± 1.57 log_10_ copies number/µg of total cDNA, respectively, which were significantly higher than the surviving shrimp in the negative control and 5–10 mg/mL Ua HWCE–YHV-injected groups with 0.00 ± 0.00, 2.57 ± 0.00, and 3.08 ± 1.52 log_10_ copies number/µg of total cDNA, respectively (*p* < 0.05) (Figure 6d1).

For the YHV load in the hepatopancreas from dead and surviving shrimp, it was shown that the viral load from the dead shrimp in the positive control group (5.27 ± 0.47 log_10_ copies number/µg of total cDNA) was significantly higher than that of the negative control (no detection) and preincubated Ui HWCE–YHV groups treated at 1–10 mg/mL, which showed YHV cDNA copy numbers of 1.77 ± 0.60, 2.55 ± 1.00, and 1.78 ± 1.48 log_10_ copies number/µg of total cDNA (*p* < 0.05) (Figure 6a2). In the Ur experiment, the viral load of all YHV-injected groups was significantly lower than that of the positive control group, except for the freshly dead shrimp of the 10 mg/mL Ur HWCE–YHV-injected group, which was 4.68 ± 0.80 log_10_ copies number/µg of total cDNA (Figure 6b2).

The viral loads of the 5 and 10 mg/mL preincubated Cl HWCE–YHV groups (5.04 ± 0.44 and 4.70 ± 0.42 log_10_ copy numbers/µg of total cDNA) were significantly higher than the 1 mg/mL preincubated Cl HWCE–YHV-injected group and the negative control group (Figure 6c2). The YHV load of the dead shrimp in the 1, 5, and 10 mg/mL Ua HWCE–YHV-injected groups contained 4.37 ± 1.21, 5.01 ± 0.79, and 4.32 ± 0.90 log_10_ copies number/µg of total cDNA, which were significantly different from those of the surviving shrimp in the 5 and 10 mg/mL Ua HWCE–YHV-injected groups and the negative control group (*p* < 0.05) (Figure 6d2).

#### 2.4.3. Histopathology

Six of the positive control, negative control, moribund, and surviving shrimp of each preincubated HWCE–WSSV injection group were subjected to histopathological analysis. It was clearly demonstrated that all moribund shrimp injected with various concentrations of each HWCE–WSSV injection clearly presented specific histopathological changes in their gills and intestines with hypertrophied nuclei and Cowdry A-type inclusion bodies, similar to the results observed in the positive control group. In contrast, the gill and intestinal tissues of shrimp from the negative control group and particularly the preincubated Ui- and Cl-HWCE–WSSV-injected groups showed normal characteristics (Figure 7).

For the histopathological analysis of the preincubated HWCE–YHV injection group, the target organs (gills and hepatopancreas) of the virus clearly exhibited pyknotic nuclei and karyorrhexis in the moribund shrimp from YHV-injected groups, similar to the positive control group (Figure 7).

## 3. Discussion

The three species of green macroalgae (*U. intestinalis*, *U. rigida*, and *C. lentillifera*) are the major types of algae produced from Thai algae farming. These Chlorophyceae species are an important source of water-soluble polysaccharides known as ulvans [13,17,25,29,30]. In this study, these algal species were used for the extraction of bioactive compounds by the hot water extraction method to investigate the efficacy of their biological properties for application in Pacific white shrimp culture.

The major components of the HWCEs from green macroalgae were identified as ulvan, those from red algae are mainly agarans and carrageenans, and those from brown algae are alginates and fucans, and the storage polysaccharide laminarin. Ulvan is a water-soluble sulfated polysaccharide extracted from the cell walls of green algae belonging to Ulvales (*Ulva* or *Enteromorpha*). It is mainly composed of rhamnose, xylose, glucuronic, and iduronic acids mostly distributed in disaccharide repeating units. The major repeating disaccharide units are formed through an l-rhamnose 3-sulfated linked to a d-guluronic acid residue (ulvanobiuronic acid unit A), l-iduronic acid residue (ulvanobiuronic acid unit B), d-xylose 4-sulfated residue (ulvabiose unit A), and d-xylose 4-sulfated residue (ulvabiose unit B). Generally, the molecular weights of ulvan vary from 189 to 8200 kD [16,25,28,31,32,33].

In the present study, three major components of HWCEs from Thai green macroalgae were measured. The sulfate contents found in our study corresponded to those previously reported for *U. rotundata* (10.3%–13.8%) *U. armoricana* (9.2%–12.5%) [34], and *U. clathrate* (11.0%) [35], but were less than those reported for *U. lactuca* (18.9%) [28], *U. intestinalis* (18.4% collected from Iran) [27], *U. pertusa* (19.7%) [36], and *U. conglobate* (23.04%–35.20%) [37]. Additionally, the uronic acid and carbohydrate contents of the HWCEs from the current study were widely varied. Based on the current information, the biochemical components may mainly depend on the macroalgae species, collection season, growing area, geographical differences, environmental factors, and extraction method [29,34,38,39].

The FTIR analysis of HWCEs of green macroalgae was similar to that reported for other *Ulva* species that normally present a strong signal at 1250 cm^−1^, corresponding to sulfate ester (S=O stretching), and two additional small signals at 850 and 790 cm^−1^, corresponding to C–O–S bending, which are typical of ulvans [25,26,27,28,29,31,32]. The signals at approximately 1635 and 1425 cm^−1^ were attributed to a carboxylic group (C=O). The signal between 1200 and 1000 cm^−1^ was dominated by sugar ring vibration overlapping with stretching vibrations of (C–OH) side groups and the glycosidic bond vibration (C–O–C). The spectra of all ulvans show strong signals at approximately 1050 cm^−1^, which are likely due to C–O stretching from the two main sugars (rhamnose and glucuronic acid) [26,40]. In this case, the signals at 1250, 850, and 790 cm^−1^ of the infrared spectra of Cl were slightly different from those in other species, which was consistent with the sulfate content in the HWCE from Cl, which was the significantly lowest. Moreover, the signals of the Cl spectra at approximately 1635, 1425, and 1050 cm^−1^ were not clearly similar to those of other species and were supported by the uronic acid content of the Cl HWCE, which was the significantly lowest among the extracts.

In the present study, the HWCEs of Ui, Ur, and Cl did not exhibit toxicity to the Pacific white shrimp after 1–10 mg/mL was intramuscularly injected into experimental shrimp, as indicated by the 100% survival. Accordingly, Alves et al. (2012) [41] demonstrated a nontoxic effect of ulvan extracted from *U. lactuca* using an In vitro cytotoxicity assay in fibroblast-like cells. Similarly, sulfated polysaccharide extracted from *U. clathrate* inhibited the Newcastle disease virus (NDV) In vitro without significant toxicity [35]. The methanolic extract of *Padina gymnospora* was not found to be cytotoxic to the cellular line of fibroblasts in L929 common mice and to the line of human ovarian carcinoma (OVCAR-3) [42,43]. In the fisheries field, the ichtyotoxic activity of extracts from 73 species of Maxican macroalgae, consisting of 19 species belonging to the Chlorophyta division, 19 to Phaeophyta, and 35 to Rhodophyta, were extracted by different solvents (ethanol, acetone, and distilled water) and tested against the goldfish *Carassius auratus*. The aqueous extracts of all Chlorophyta were non-toxic, and the goldfish did not display any changes, including rapid movements, stress, change in coloring, surface gasping, slow swimming, loss of equilibrium, or death within 2 h [44]. Similarly, Targett and Mitsui (1979) [45] tested the toxicity of aqueous extracts of six species of Chlorophyta found in south Florida. The aqueous extracts from these macroalgae did not show lethal toxicity to the spotfin mojarra (*Eucinostomus argenteus*). Likewise, the toxicity of the green algae *Caulerpa taxifolia* was evaluated in three models, including mice (lethality), mammalian cell cultures, and sea urchin eggs (disturbance of cell proliferation). The aqueous extracts were only active against fibroblasts and mice but not toxic to sea urchin eggs [46]. Furthermore, none of the crude extracts of four green marine algae (*Cladophora rupestris*, *Codium fragile*, *Ulva intestinalis*, and *Ulva lactuca*) appeared to possess toxicity toward mammalian L6 cells (rat skeletal muscle myoblasts) at high concentrations [47]. In contrast, the water extracts of *Stokeyia indica* and *Caulerpa racemosa* exhibited the most cytotoxic activity (LC_50_ value below 70 mg/mL) in our brine shrimp (*Artemia salina*) lethality assay [48]. It is possible to infer that ulvans or other components derived from HWCEs of microgreen algae are relatively nontoxic. In addition, the sulfate polysaccharides from these algae have, among other characteristics, low toxicity and, as a result, cause few adverse effects [40].

Antibacterial activity against *V. alginolyticus*, *V. vulnificus*, *V. harveyi*, *V. parahaemolyticus*, and *V. parahaemolyticus* AHPND was not observed for the HWCEs from the Ui, Ur, and Cl and commercial ulvan from Ua because all concentrations of the tested substances incompletely inhibited the growth of all tested bacteria. Although slight inhibition was observed in the Ui and Cl experiments, *V. vulnificus* was inhibited only for a short period from 6–18 h after exposure. Based on these data, MIC assays failed to identify the efficacy of the tested substances, indicating that the HWCEs from the three selected green macroalgae were not effective in controlling pathogenic *Vibrio* spp. However, the application of these substances inversely promoted *Vibrio* growth, particularly for *V. parahaemolyticus* and *V. parahaemolyticus* AHPND, as indicated by the rapid increase in their absorbance values. Thus, these HWCEs are not applicable for controlling *V. parahaemolyticus* and may contain various types of monosaccharides, which act as nutrient sources for *Vibrio* spp. Likewise, Bansemir et al. (2006) [49] reported that dichloromethane, methanol, and water extracts of 26 species of cultivated macroalgae (19 Rhodophyta, three Phaeophyta, and four Chlorophyta) were screened for their antibacterial activity against fish pathogenic bacteria. Only the dichloromethane extracts showed significant inhibitory effects. The methanol extracts showed moderate or weak antimicrobial activity, and the water extracts showed no antibacterial activity. On the other hand, Alghazeer et al. (2013) [50] described the antibacterial activity of the crude methanol and water extracts of 19 marine algae species composed of six Chlorophyta, eight Phaeophyta, and five Rhodophyta against four Gram-positive bacteria, *Staphylococcus aureus*, *Bacillus subtilis*, *Bacillus* sp., and *S. epidermidis,* and four Gram-negative bacteria, *Escherichia coli*, *Salmonella typhi*, *Klebsiella* spp., and *Pseudomonas aeruginosa*. The results showed significant antibacterial activity against Gram-positive and Gram-negative bacteria. The methanol extract of *Caulerpa racemosa* exhibited strong inhibition of *Klebsiella* spp. and *S. typhi,* with significantly higher inhibition than all other algae methanol extracts (*p* < 0.05); moreover, most green algae aqueous extracts of *Enteromorpha* spp. and *E. prolifera* showed distinct antibacterial activity against *Bacillus* spp., *S. epidermidis*, and *Klebsiella* spp. Similarly, *Ulva flexuosa* aqueous extract displayed strong antagonistic activity against Gram-positive/negative bacteria (*Bacillus subtilis*, *B. pumulis*, *Enterococcus faecalis*, *Staphylococcus aureus*, *S. epidermidis*, *Escherichia coli*, *Klebsiella pneumoniae*, *Pseudomonas aeruginosa*) [51].

However, research recently intensified on the crucial utilization of seaweeds and polysaccharides derived from algae as therapeutic agents to control diseases caused by *Vibrio* spp. and other bacteria in shrimp or other animals [52,53]. For example, the polysaccharide extracts from *C. racemosa* showed antibacterial activity against *E. coli*, *Streptococcus aureus*, and *Salmonella* sp., the extracts of *Enteromorpha linza* inhibited *E. coli*, *S. typhimurium**,* and a polysaccharide extracted from *Chaetomorpha aerea* was active against *B. subtilis*, *Micrococcus luteus*, and *S. aureus* [54]. Similarly, the extracts from *C. sertularioides* using five different solvents, including acetone, chloroform, ethanol, ethyl acetate, and methanol, showed significant antibacterial activity against six bacterial pathogens [55]. In addition, the ethanol extracts of *Ulva pertusa*, *U. prolifera*, *Gloiopeltis furcata*, *Gracilariopsis lemaneiformis*, *Sargassum fusiforme*, and *Ishige okamurae* showed antibacterial activity against *S. aureus*, *A. hydrophila*, and *G. lemaneiformis*, while *U. prolifera* displayed the greatest inhibition of *A. hydrophila*. Only *S. fusiforme* showed antibacterial activity against *P. aeruginosa* and against four out of the five species of bacteria [56]. Miranda et al. (2016) showed that methanol macroalgae extracts from *C. sertularioides* and *U. lactuca* might be possible alternatives for the prevention of vibriosis in *L. vannamei* [57], and Selvin et al. (2011) showed that dietary addition of dichloromethane–methanol (1:1) extracts from the marine green algae *U. fasciata* was highly effective for controlling bacterial pathogens (*V. fischeri*, *V. alginolyticus*, *V. harveyi*, and *Aeromonas* sp.) in shrimp [7]. The exact antibacterial mechanisms and structural relationships between sulfated polysaccharides and the cell membrane receptors of pathogenic bacteria are not yet elucidated. Nevertheless, one possible mechanism of the antibacterial efficacy of sulfated polysaccharides is the binding of these materials to the bacterial cell surface and subsequent damage to the cell membrane, which leads to the leakage of essential nutrients, resulting in the death of the bacteria [53,58,59,60].

Based on information from this study and previous reports, it was suggested that different extraction methods and various types of macroalgae may provide different bioactive compounds, which can further alter the effective concentrations for controlling microorganisms. Additionally, the different targets of microbial agents may also be one of the key factors in their ability to damage cell membranes or their cellular components. Further study is needed to clarify these hypotheses.

The anti-WSSV activity of the HWCEs from Ui, Ur, and Cl in Pacific white shrimp was manifested as significantly lower cumulative mortality in pre-incubated Ui- and Cl-HWCE–WSSV-injected shrimp, and these findings were supported by significantly lower viral loads in the gills and intestine by qPCR and histological analyses. It could be summarized that the HWCEs from Ui and Cl were very effective in controlling WSSV. However, with careful examination of the qPCR results in both the gills and intestine, high and low viral loads were still found in the shrimp intestine of surviving shrimp co-injected with 1 mg/mL of Ui HWCE–WSSV or 10 mg/mL of Cl HWCE–WSSV solutions. This suggests that only 5 and 10 mg/mL Ui HWCEs provide adequate protection against WSSV, while the effects of 1–10 mg/mL Cl HWCE still need further investigation to support its efficacy in inactivating WSSV.

Regarding the anti-YHV activity of Ui, Ur, and Cl HWCEs in Pacific white shrimp, the cumulative mortality of the Ui HWCE–YHV-injected group was significantly lower than the viral load in the gills and hepatopancreas using qRT-PCR. Although the Ui HWCE–YHV-injected groups showed a good survival rate, the viral load in both the gills and hepatopancreas of these groups was not significantly different from that in the positive YHV-injected group. This result suggests that 1–10 mg/mL Ui HWCEs were not perfect in controlling YHV infection. The YHV particles may have been inhibited but can survive at high doses without mortality. These interesting results require further clarification.

The effective mechanisms of green macroalgae HWCEs in WSSV and YHV inactivation are poorly understood. The information obtained from relevant research was reviewed. A study of the antiviral properties of algal extracts, particularly against WSSV, by oral administration via enriched *Artemia nauplii* was conducted. Algal extract concentrations of 400 to 750 mg/L were used to enrich *Artemia*, which were continuously fed to shrimp for 20 days. It was found that this application method is an effective strategy for protection against WSSV [61,62,63]. In these works, the key bioactive sulfate polysaccharide components, particularly fucoidans from brown algae and sodium alginates from red algae, were shown to function in virus control [22,64,65,66].

The direct properties of the HWCEs and the polysaccharides extracted from green algae were not previously investigated. However, the extract “ulvan” from *Enteromorpha* (*Ulva*) *intestinalis* acts as an immunostimulant against WSSV in juvenile black tiger shrimp (*Penaeus monodon*) by activating cellular immunity [67]. Additionally, Lauzon et al. (2015) [68] showed that ulvan extracted from *E. intestinalis* enhances immune responses in *L. vannamei* and *P. monodon* juveniles when administered via incorporation into the diet. It was found that this ulvan extract was very effective against WSSV. However, in our present study, the effectiveness of a commercial ulvan against WSSV was not sufficient; 1 and 5 mg/mL Ua adequately protected shrimp from viral infection, while 10 mg/mL caused WSSV infection to be more severe. All surviving or dead shrimp from these groups still showed high viral loads in both the gills and intestine. In contrast, the commercial ulvan did not have the ability to protect shrimp from YHV infection because all concentrations of preincubated Ua HWCE–YHV-injected groups showed the highest amount of dead shrimp and viral loads in the gills and hepatopancreas among treatment groups. These results suggest that ulvan may not be a key component to protect shrimp from WSSV or YHV.

The virus-inhibiting effect of sulfate polysaccharides appears to be mainly based on their ability to interfere with the initial attachment of the enveloped virus to the target cells via the ionic interaction between positively charged regions of the external viral glycoprotein and the negatively charged constituents of the cell surface, consequently leading to the blockage of viral entry [69]. The structural diversity and complexity of algal polysaccharides and their derivatives contribute to their antiviral activities in different phases of many different viral infection processes, including inhibition of viral attachment, inhibition of viral internalization and uncoating, and inhibition of viral transcription and replication [16,70].

Aguilar-Briseño et al. (2015) [35] supported the concept that ulvan and fucoidan also have the same mechanism of antiviral activity and that ulvan inhibits viral fusion by interacting with the intact F0 protein, preventing it from being cleaved into the mature form. Ulvan alone showed better anti cell–cell spread activity than fucoidan, but both showed even stronger effects with used in combination. Interestingly, Sotanon et al. (2018) [71] recently found that the binding of an envelope protein of the WSSV virion, namely, VP37, to shrimp hemocytes, was impaired by preincubation of VP37 with sulfated galactan, a sulfated polysaccharide derived from red algae (*Gracillaria fisheri*), and this is consistent with the report of Wongprasert et al. (2014) [22], who discovered that shrimp were effectively protected from WSSV infection by sulfated galactan. Sotanon et al. (2018) [71] also demonstrated that VP37 might be able to recognize sulfated galactan as its binding partner and that the binding of VP37 to shrimp hemocytes might target a molecule whose structure is related to sulfated galactan. For this reason, it is possible that the VP37 of WSSV can bind with the ulvan from green algae because sulfated galactan and ulvan are the same sulfated polysaccharide in marine algae. YHV is also an envelope virus similar to WSSV. It is possible that YHV may contain some envelope proteins, which effectively bind with ulvan, making it difficult for viral particles to infect the shrimp cells.

In comparison with our current results, however, the antiviral activities of HWCEs of Ui, Ur, and Cl were not consistent with the previous hypotheses because the negative charge (shown in the sulfate and uronic acid contents) of the Ur HWCE was higher than the crude polysaccharide from Ui and Cl. Additionally, the cumulative mortality and viral load of the preincubated Ur HWCE–WSSV-injected group were significantly higher than those of the other groups. Further research is needed to clarify these little-known mechanisms.

## 4. Materials and Methods

### 4.1. Extraction of Hot Water Crude Extracts (HWCEs)

Three local species of Thai green algae, including *U. intestinalis* (Ui), *U. rigida* (Ur), and *C. lentillifera* (Cl), were harvested from earthen ponds in Phetchaburi Coastal Aquaculture Research and Development Center in Phetchaburi Province, Thailand. The green algae were washed with tap water to remove all mud and epiphytes. The fleshy green algae were cut to a smaller size. The crude polysaccharide was extracted by the hot water extraction method, and approximately 100 g of algae sample was boiled in 500 mL of distilled water (1:5 *w*/*v*) for 90 min at 110 °C. The hot extract was filtered from solid residue. The supernatant was condensed to a minimal volume using a rotary evaporator (BÜCHI Rotavapor R 200). Dialysis against distilled water was performed to remove excess salts using dialysis tubing (6–8000 Da molecular weight cut off), and the samples were then freeze-dried.

### 4.2. Chemical Analyses

The sulfate content was measured after acid hydrolysis (2 N HCL at 100 °C for 2 h) of the polysaccharides, according to the BaCl_2_–gelatin method, using K_2_SO_4_ as a standard [72]. The carbohydrate content was determined by the phenol-sulfuric method using glucose as a standard [73]. The uronic acid content was measured by following the carbozole method using d-glucuronic acid as a standard [74].

### 4.3. Structure of HWCEs

The qualitative analysis of the active components of the HWCEs was analyzed by Fourier-transform infrared (FTIR) spectroscopy (PERKIN ELMER Spectrum One, Waltham, MA, USA). HWCEs were mixed with KBr to make a transparent film. The frequency of the spectra for analysis was between 4000 and 400 cm^−1^ wavelengths, and the vibration spectra were recorded graphically.

### 4.4. Experimental Animals

In total, 1600 healthy Pacific white shrimps free from bacterial and viral infections (22.57 ± 2.41 g) were collected from Phetchaburi Coastal Aquaculture Research and Development Center, Phetchaburi Province, Thailand; they were acclimatized in a 3000-L polyethylene tank with 20 ± 2 ppt seawater in an aeration system for seven days. During acclimatization before the start of the experiments, shrimp were fed three times per day with artificial feed at 5% of their body weight. Ten shrimp were randomly sacrificed, and DNA and RNA of each shrimp were extracted and used for WSSV and YHV detection with the same methods as described below, while VP_AHPND_ was also carried out with the method of Reference [8] in order to confirm pathogen-free conditions. 

### 4.5. Lethal Dose Analysis of HWCEs

The HWCEs from local Thai green algae, including *U. intestinalis* (Ui), *U. rigida* (Ur), *C. lentillifera* (Cl), and a commercial purified ulvan from *U. armoricana* (Ua) (Elicity, Crolles, France), were administered the acute toxicity dose. The HWCE test concentrations were 1, 5, and 10 mg/mL. In total, 360 shrimp from Section 4.4 were separated into four groups, with 10 shrimp per group per concentration (in triplicate), and injected intramuscularly with 0.1 mL of each concentration of HWCE. The mortality rates were recorded every hour for the first 6 h and every 6 h for seven days. 

### 4.6. Antibacterial Activity of HWCEs from Local Thai Green Algae

#### 4.6.1. Preparation of *Vibrio* spp.

Pathogenic strains of *Vibrio* spp. (*V. alginolyticus* (VAAAHM01), *V. vulnificus* (VVAAHM01), *V. harveyi* (VHAAHM01), *V. parahaemolyticus* (VPAAHM01), and *V. parahaemolyticus* AHPND strain (VPAAHM01_AHPND_)) were isolated from diseased Pacific white shrimp from a private farm in southern Thailand. The *Vibrio* spp. were stored at −80 °C in the Laboratory of Aquatic Animal Health Management (AAHM), Department of Aquaculture, Faculty of Fisheries, Kasetsart University. A single colony of *Vibrio* spp. was cultured in tryptic soy broth (TSB, Difco, MD, USA) and incubated in an air shaker at 30 °C for 18 h. The bacterial suspensions were centrifuged at 1500 rpm for 15 min. Bacterial pellets were washed twice and resuspended in sterile 1.5% NaCl. The concentration of *Vibrio* spp. was adjusted to 1 × 10^6^ colony-forming units (cfu)/mL for the minimal inhibitory concentration (MIC) tests.

#### 4.6.2. Minimal Inhibitory Concentrations (MICs) by Liquid Growth Inhibition Methods

The antibacterial activity of crude polysaccharide from local Thai green algae was determined by the MICs with the liquid growth inhibition method in a 96-well microtiter plate (in triplicate) [75]. The final concentrations of each crude polysaccharide were 0.0, 0.5, 1.0, 2.0, 4.0, 8.0, 16.0, 32.0, 64.0, and 128.0 μg/mL, and the bacterial concentration was 1 × 10^5^ cfu/mL. Enrofloxacin (128.0 µg/mL) was used as a control, since it was reported as the most potent drug to *Vibrio* spp. [76]). The growth of bacteria in each well was measured at 600 nm in an iMark^TM^ Microplate Absorbance Reader (Bio-Rad, Hercules, CA, USA) at 0, 1, 3, 6, 9, 12, 18, 24, and 48 h after incubation at 30 °C.

### 4.7. Antiviral Activity of HWCEs from Local Thai Green Algae 

#### 4.7.1. Experimental Animals

In total, 1200 shrimp from Section 4.4 were randomly selected and maintained under the same conditions as described in Section 4.4.

#### 4.7.2. Preparation of WSSV and YHV Inoculum

WSSV and YHV inocula were prepared from WSSV- and YHV-infected shrimp tissue following the procedure of Nunan et al. [77]. Head soft tissue from the cephalothorax, including the gills, was homogenized in TN buffer (0.02 M Tris-HCl, 0.4 M NaCl, pH 7.4) using a hand blender and centrifuged at 1000× *g* for 10 min at 4 °C. The final supernatant was filtered through a 0.45-µm membrane filter and stored at −80°C. The presence of WSSV and YHV in the tissue sample of the infected shrimp was checked by PCR for WSSV and RT-PCR for YHV [22,78].

#### 4.7.3. Median Lethal Dose Assay at 96 H (96-H LD_50_)

In total, 420 shrimps among those described in Section 4.7.1 were injected (intramuscular) with 100 µL of a 10-fold serial dilution of WSSV and YHV in seven dilutions (10^−4^–10^−10^). Thirty shrimp were used per dilution per pathogen. Moribund and dead shrimp were recorded, removed from the aquarium and processed for detection of viral infection. Surviving shrimp were collected at the end of the experiment (96 h post injection) and analyzed for viral infection [22,78]. The median lethal dose (96-h LD_50_) was calculated by determining the 50% endpoint titer using the following formula [79]:$$\mathrm{Death}\;\mathrm{score}\;=\frac{\mathrm{number}\;\mathrm{of}\;\mathrm{dead}\;\mathrm{animals}\;\mathrm{per}\;\mathrm{dilution}}{\mathrm{number}\;\mathrm{of}\;\mathrm{animals}\;\mathrm{inoculated}\;\mathrm{per}\;\mathrm{dilution}}.$$
log10 50% End point dilution =−(total death score+0.5)×log dilution factor.
$$50\%\;\mathrm{End}\;\mathrm{point}\;\mathrm{dilution}\;=10^{\log10\;50\%\;\mathrm{End}\;\mathrm{point}\;\mathrm{dilution}}.$$
$$\mathrm{The}\;\mathrm{titer}\;\mathrm{of}\;\mathrm{the}\;\mathrm{virus}\;=10^{-(\log10\;50\%\;\mathrm{End}\;\mathrm{point}\;\mathrm{dilution})}({\mathrm{LD}}_{50}/\mathrm{mL}).$$

#### 4.7.4. Antiviral Activity Analysis

In vitro antiviral activity against WSSV and YHV was assessed following the methods of Balasubramanian et al. and Palanikumar et al. [80,81]. The concentration of both viral suspensions was calculated from a 50% end point dilution that was mixed with HWCEs at various concentrations: 1, 5, and 10 mg/mL. The positive control consisted of a mixture of both viral suspensions and a phosphate-buffered saline (PBS buffer, pH 7.4) in a ratio of 1:1, and the negative control was PBS buffer. All these mixtures were incubated at 30 °C for 2 h before injection. After 2 h, the preincubated solution was injected intramuscularly into experimental shrimp in triplicate (*n* = 10 × 3 = 30). Mortalities were recorded each day. The dead and surviving shrimp were collected from the target organs of WSSV infection (gills and intestine) and YHV infection (gills and hepatopancreas) for the measurement of viral loads and histopathological analysis.

#### 4.7.5. Investigation of Viral Load in Target Tissues Using Quantitative Real-Time PCR for WSSV and Quantitative Real-Time RT-PCR for YHV

Gills and intestine from dead and surviving shrimp of each experimental group (*N* = 6) were collected, and the total genomic DNA and associated WSSV viral DNA were extracted using DNAzol reagent (Molecular Research Center, Inc., Cincinnati, OH, USA) according to the manufacturer’s instructions. The viral load was estimated by absolute quantification using the standard curve method. One hundred nanograms of total genomic DNA and associated WSSV viral DNA was amplified with an AriaMx Real-time PCR System (Agilent, Santa Clara, CA, USA) and Agilent Brilliant III Ultra-Fast SYBR qPCR MM (Agilent, Santa Clara, CA, USA) according to the manufacturer’s recommended protocol using the WSSV VP28 specific primers (forward primer 5’ TGTGACCAAGACCATCGAAA 3’ and reverse primer 5’ ATTGCGGATCTTGATTTTGC 3’) to amplify a 161-bp fragment of the VP28 gene of WSSV [21].

For YHV infection, the target organs of YHV infection (gills and hepatopancreas) were extracted from the total RNA using RNAzol reagent (Macherey-Nagel company, Düren, Germany), and cDNA was synthesized using the ReverTra Ace qPCR RT Master Mix Kit (Toyobo, Osaka, Japan) according to the manufacturer’s instructions. The viral load was estimated by absolute quantification using the standard curve method. One hundred nanograms of cDNA and associated YHV viral cDNA were amplified with an AriaMx Real-time PCR System (Agilent, Santa Clara, CA, USA) and Agilent Brilliant III Ultra-Fast SYBR qPCR MM (Agilent, Santa Clara, CA, USA) according to the manufacturer’s recommended protocol using the YHV specific primers (forward primer 5’ CCGCTAATTTCAAAAACTACG 3’ and reverse primer 5’ AAGGTGTTATGTCGAGGAAGT 3’) to amplify a 135-bp amplicon [76].

#### 4.7.6. Histopathological Analysis

One centimeter from the cephalothorax of moribund and surviving shrimp from each treatment was collected and immediately fixed with Davidson’s solution for 24–48 h. Tissues were dehydrated in alcohol (30%–100%) and embedded in paraffin blocks. Sections 5 µm in thickness were stained with hematoxylin and eosin (H&E) and examined under a light microscope.

### 4.8. Statistical Analysis

The data obtained in this study were presented as the means ± SD and analyzed by one-way analysis of variance (ANOVA). Duncan’s new multiple-range test (DMRT) was used to test for significant differences among groups. Statistical significance was set at a *p*-value less than 0.05.

## 5. Conclusions

In conclusion, the present study demonstrated that hot water extraction is an effective method for sulfate polysaccharide extraction, as indicated by chemical and structural analyses of crude polysaccharides from three species of green algae compared with commercial purified ulvan. The HWCEs from three species of green macroalgae were a safe bioactive product for use with Pacific white shrimp. Furthermore, the results of this study showed that the HWCEs from *U. intestinalis* were able to strongly inhibit WSSV and YHV infection but were not active in controlling shrimp pathogenic bacteria (*Vibrio* spp.). The ability of the HWCEs of three species of green algae to protect against viral infection should be investigated for other virulent viruses, namely, TSV and SHIV, which also cause mass mortality in marine shrimp. Moreover, the effects of the HWCEs from *U. intestinalis* as a feed additive on the immune responses and disease resistance in Pacific white shrimp should be further investigated to confirm the exact performance of these compounds and to generate a new practical method of disease prevention in the shrimp aquaculture industry.

## Figures and Tables

**Figure 1 marinedrugs-18-00140-f001:**
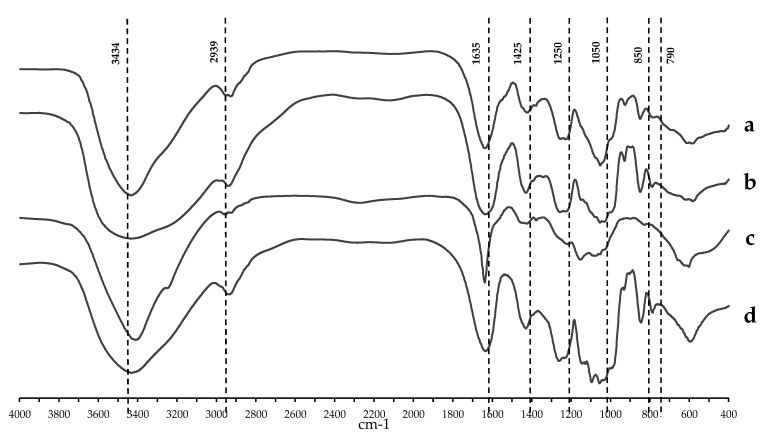
The structure of the hot water crude extracts (HWCEs) from three species of local Thai green macroalgae analyzed by Fourier-transform infrared (FTIR) spectroscopy: (**a**) *U. intestinalis*, (**b**) *U. rigida*, (**c**) *C. lentillifera*, and (**d**) a commercial ulvan from *U. amoricana*.

**Figure 2 marinedrugs-18-00140-f002:**
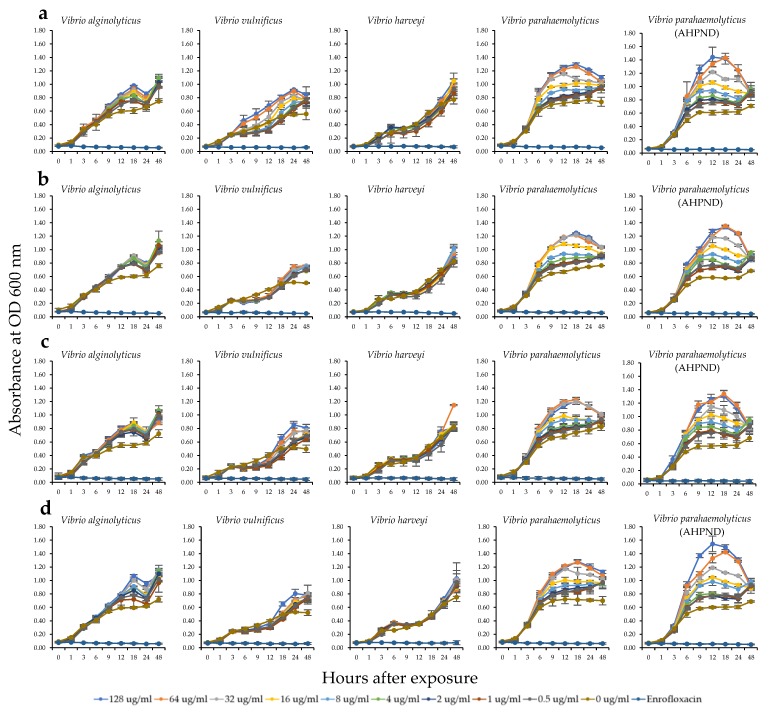
Antibacterial activity of HWCEs from three species of green algae: *U. intestinalis* (**a**), *U. rigida* (**b**), *C. lentillifera* (**c**), and a commercial ulvan (**d**) from *U. armoricana* against *Vibrio alginolyticus*, *V. vulnificus*, *V. harveyi*, *V. parahaemolyticus*, and *V. parahaemolyticus* AHPND strain.

**Figure 3 marinedrugs-18-00140-f003:**
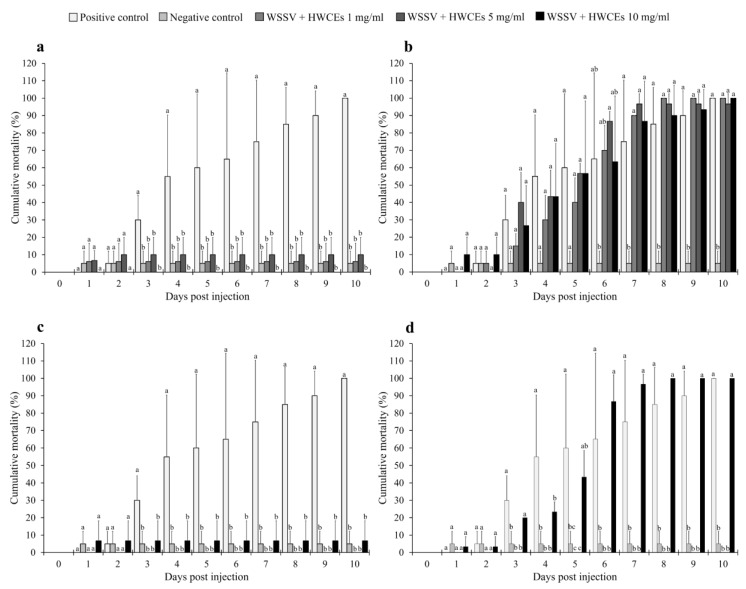
Cumulative mortality of Pacific white shrimp post injection with preincubated HWCE (from Ui (**a**), Ur (**b**), Cl (**c**), and Ua (**d**) at 1, 5, and 10 mg/mL) white spot syndrome virus (WSSV) solutions. Different letters on the bars at different days indicate significant differences (*p* < 0.05).

**Figure 4 marinedrugs-18-00140-f004:**
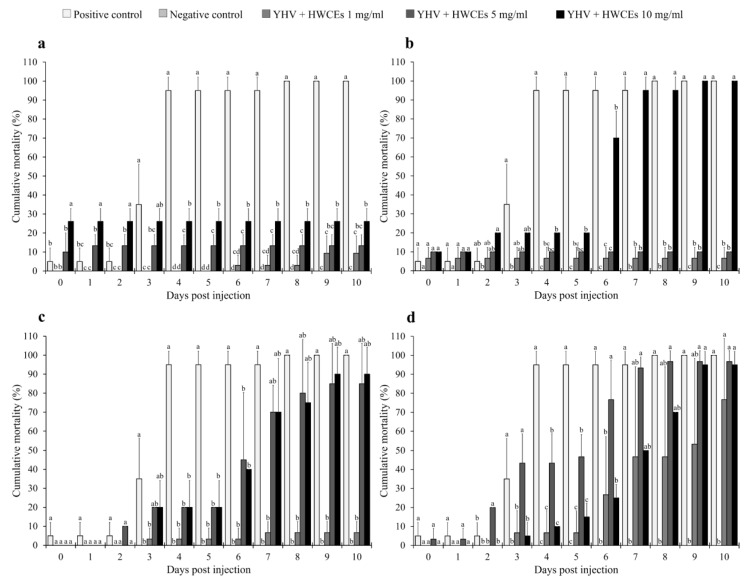
Cumulative mortality of Pacific white shrimp post injection with preincubated HWCE (from Ui (**a**), Ur (**b**), Cl (**c**), and Ua (**d**) at 1, 5, and 10 mg/mL) yellow head virus (YHV) solutions. Different letters on the bars at different days indicate significant differences (*p* < 0.05).

**Figure 5 marinedrugs-18-00140-f005:**
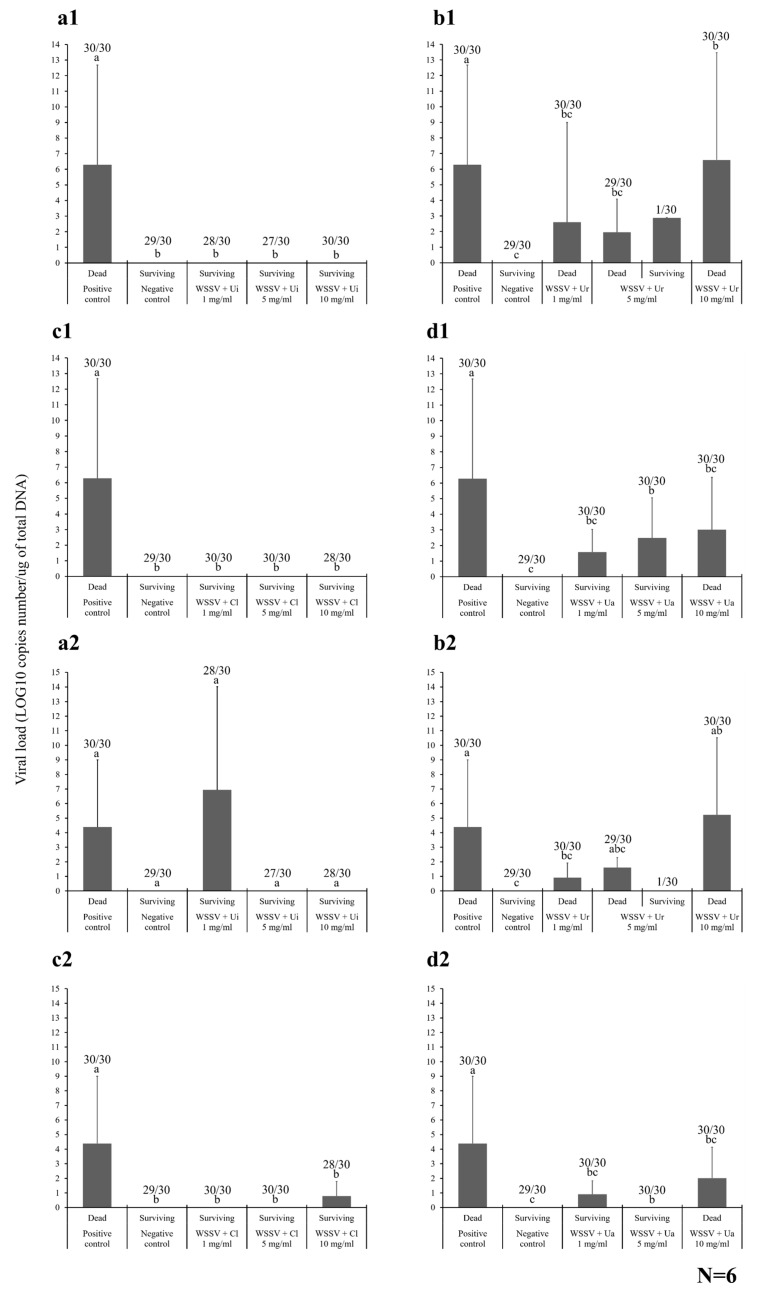
Investigation of the WSSV load in the gills (**a1**, **b1**, **c1,** and **d1**) and intestines (**a2**, **b2**, **c2,** and **d2**) of Pacific white shrimp after injection with preincubated HWCEs (from Ui (**a**), Ur (**b**), Cl (**c**)) WSSV solutions and preincubated WSSV–commercial ulvan from Ua (**d**), at 1, 5, and 10 mg/mL. Different letters on the bars in different groups indicate significant differences (*p* < 0.05).

**Figure 6 marinedrugs-18-00140-f006:**
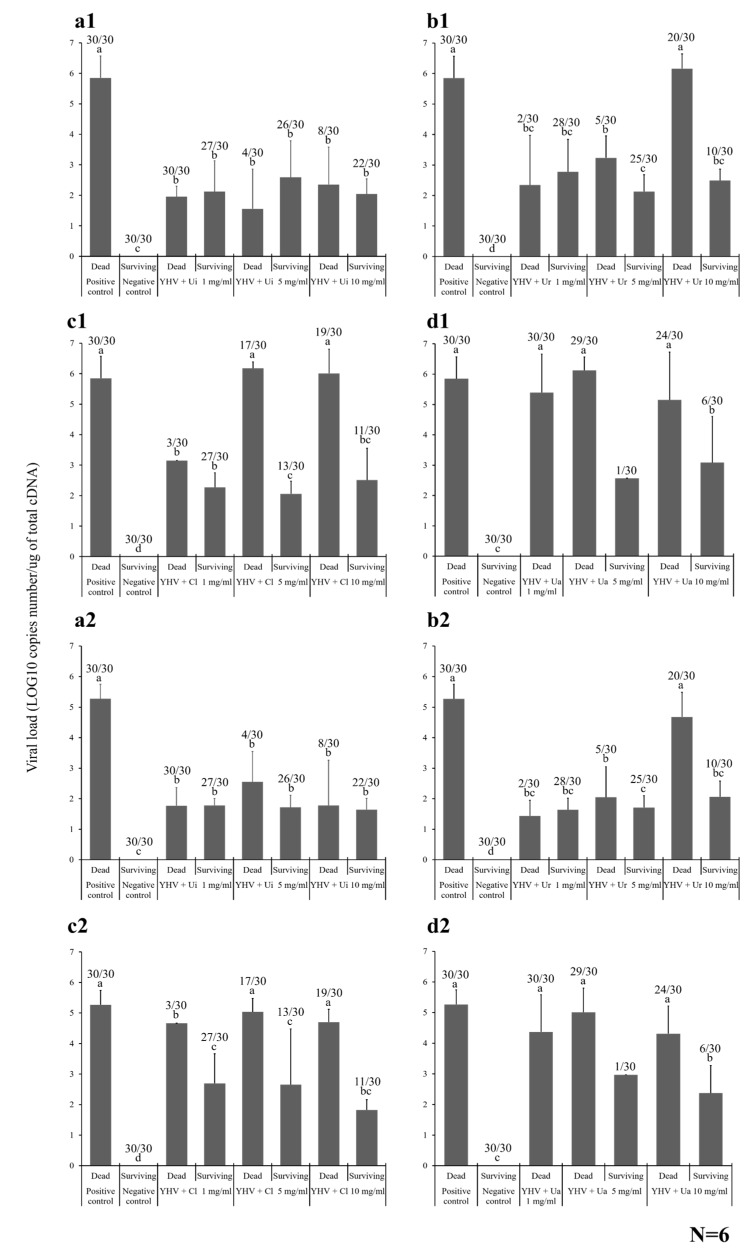
Investigation of the YHV load in the gills (**a1**, **b1**, **c1,** and **d1**) and hepatopancreas (**a2**, **b2**, **c2,** and **d2**) of Pacific white shrimp after injection with preincubated HWCEs (from Ui (**a**), Ur (**b**), Cl (**c**)) -YHV solutions and a preincubated YHV–commercial ulvan from Ua (**d**) at 1, 5, and 10 mg/mL. Different letters on the bars in different groups indicate significant differences (*p* < 0.05).

**Figure 7 marinedrugs-18-00140-f007:**
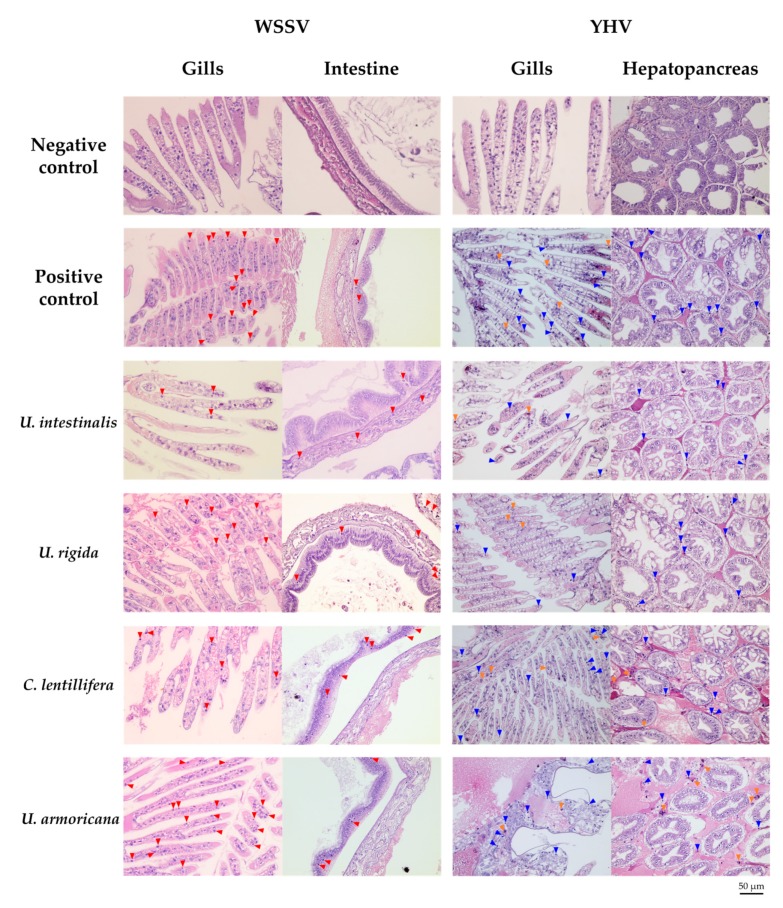
Histopathology of normal, WSSV- and YHV-infected Pacific white shrimp on the target tissues (the gills, intestine and hepatopancreas). The normal shrimp selected from negative control, and moribund shrimp selected from the positive control, HWCE–WSSV-injected shrimp, and HWCE–YHV-injected shrimp. Red arrows indicate nuclear hypertrophy. Blue and orange arrows indicate pyknotic nuclei and karyorrhexis, respectively.

**Table 1 marinedrugs-18-00140-t001:** Chemical composition of local Thai green macroalgae crude extracts and a commercial ulvan from *Ulva armoricana*. Different letters in the same column indicate significant differences (*p* < 0.05).

Source of crude extract	Sulfate content (%)	Uronic acid content (%)	Carbohydrate content (%)
*U. intestinalis* (Ui)	11.01 ± 0.29^b^	29.58 ± 1.66^b^	50.10 ± 9.18^b^
*U. rigida* (Ur)	13.89 ± 0.38^a^	32.76 ± 1.53^b^	51.02 ± 3.72^b^
*Caulopa lentillifera* (Cl)	7.29 ± 0.28^c^	9.22 ± 0.56^c^	6.85 ± 1.74^c^
*U. armoricana* (Ua)	11.50 ± 0.44^b^	41.47 ± 4.98^a^	64.03 ± 2.75^a^

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
