# Peer review of "Antibacterial and Antiviral Activities of Local Thai Green Macroalgae Crude Extracts in Pacific White Shrimp (Litopenaeus vannamei)"

_marinedrugs, 2020, doi:10.3390/md18030140_

Round 1
Reviewer 1 Report
The authors should report the detailed methodology followed during experimentation.
missing references lines 41-42; 49-50; 50-51; 68-69; 90-91; 91-92; 95-97 L45-46 mentions clinical signs of wssv. More signs need to be given that are cardinal to the disease A detailed methodology of how they generated their results needs to be shared. This is important for reproducibility purposes.Author Response
Reviewer 1
The authors should report the detailed methodology followed during experimentation.
missing references lines 41-42; 49-50; 50-51; 68-69; 90-91; 91-92; 95-97 L45-46 mentions clinical signs of WSSV. More signs need to be given that are cardinal to the disease. A detailed methodology of how they generated their results needs to be shared. This is important for reproducibility purposes.
Response: Thank you for your comments.
1) We have already added the references in line 41-42; 49-50; 50-51; 68-69 and we already added the gross signs of white spot syndrome in line 45-46 with a proper reference [4].
2) In the current version of the manuscript, we put references [20] and [23] 92-94 and 94-95, respectively. But in line 95-97, it could not be edited because this sentence includes our knowledge background and opinions.
3) In order to catch up with the standard scientific writing, our methodology was delicately drawn based on a standard protocol, which may ease to reproduce. Some information could not be narrated for more details; however, some proper references were indicated for specific following. For example, “Methodology of antiviral activity”, we applied the methods from Balasubramanian et al. (2007) and Palanikumar et al. (2018) as references “[80,81]” and we have already referenced in subsection 4.7.4.
Reviewer 2 Report
The paper deals with an interesting topic and fits the scopes of the journal. However, it cannot be accepted for publication due to some major concerns.
The first one is that pathogens were preincubated with the extracts and then injected into the shrimp. The in vitro effects were not tested before the injection, the possible direct effects of the extracts on the viability of them is not provided.
Introduction: First viruses and then bacteria and in mat and met and results just the oposite. Please, be consistent.
Materials & methods:
Some more important data should be included. For example , the strain of bacteria used or how if was demonstrate that the White shrimp were free of bacterial and viral infections? (line 129). Why wo strains of V. parahaemolyticus were included?. Add this info.
Eurofloxacin was a control of what?. Discuss this point.
There are two subsections with animals. A great amount is included but nota ll of them used. Please clarify this aspect. The reduction of animals in research shoudl be priority.
Lines 131-132. Confusing text, were moribund shrimps or not?. Please, clarify.
Virus titre shoudl be done, not only presence.
Statistic study should be repeat. Please indicate clearly if a doublé ANOVA was done or why the Duncan test was selected.
Results. Bacteria grows more with the extracts (line 149). Discuss this.
“comercial ulvan … a strange results” (line 187), please explain properly.
Figure legends. Explained a, b and c fig but not the d fig. Legends in fig 3 and 4 identical.
Cannibalism is indicated as probably (l 244). How can this fact affect the results?
Cuantification of detected damages should be done by image analysis. Histopathology is of 2 groups and there are 4 in results (line 321).
Discussion: Why were selected these HWCEs concentrations or such incubations times?
Line 580 “ as described in 4.4.” and it was not described.
Author Response
Reviewer 2
The paper deals with an interesting topic and fits the scopes of the journal. However, it cannot be accepted for publication due to some major concerns.
Response: thank you for your comments.
1) The first one is that pathogens were preincubated with the extracts and then injected into the shrimp. The in vitro effects were not tested before the injection, the possible direct effects of the extracts on the viability of them is not provided.
Response: In terms of application, safety or toxicity of the target substances to be applied is the first priority of the standard protocol. In the manuscript, therefore, we have already determined the acute toxicity of HWCEs in section 4.5 and the results of acute toxicity (section 2.2) showed that the HWECs have no effects to tested shrimp and all tested experimental animals of this part are survived.
2) Introduction: First viruses and then bacteria and in mat and met and results just the opposite. Please, be consistent.
Response: In order to increase the better flow of our manuscript, we have to firstly demonstrate the results obtained from bacteria, which showed slightly effects. Then, we moved further to narrate and highlight all obviously results found in viral challenging experiments.
3) Materials & Methods:
Some more important data should be included. For example, the strain of bacteria used or how if was demonstrate that the White shrimp were free of bacterial and viral infections? (line 129). Why two strains of V. parahaemolyticus were included? Add this info.
Response:
3.1) The type strains of each bacteria used were indicated in the manuscript (lines 587-589).
3.2) The methods used to declare about no contamination of target pathogens were described in section 4.4 (Lines 577-579).
3.3) We chose two strains of V. parahaemolyticus for testing the antibacterial activity because they can seriously cause vibriosis disease in the shrimp culture industry with different manners.
4) Eurofloxacin was a control of what? Discuss this point.
Response: We added and indicated this point in the section 4.6.2 (line 603) and placed a reference [76] of Roque et. al., 2001 there.
5) There are two subsections with animals. A great amount is included but not all of them used. Please clarify this aspect. The reduction of animals in research should be priority.
Response: We had hardly tried to minimize the number of experimental shrimps based on the 3R protocol of animal ethics. But we had to stock more shrimp because in shrimp they harmfully possess cannibalism behavior to kill each one another. Then the heathier shrimp and survived shrimp were further used to carry out each target experiment.
6) Lines 131-132. Confusing text, were moribund shrimps or not? Please, clarify.
Response: All shrimp in section 2.2 are survived and healthy. We indicated this matter so as to confirm the health status of experimental shrimp used to obtain all those data.
7) Virus titer should be done, not only presence.
Response: We had already conducted and calculated the titer of the viruses following the method described in subsection 4.7.3. The results of the titer of WSSV and YHV are shown in the result in subsection 2.4.1, 107.75 LD50/ml at 96 hours and 109.52 LD50/ml at 96 hours respectively.
8) Statistic study should be repeat. Please indicate clearly if a doublé ANOVA was done or why the Duncan test was selected.
Response: Statistically, ANOVA is basically used to test the hypothesis of the target research relying on significant or non-significant differences among treatments. DMRT or any mean comparison methods, such as LSD, Turkey’s test, ect., is further used to test mean significant differences between each treatment pair. Therefore, our statistical analyses in the manuscript are correct methods.
Results. Bacteria grows more with the extracts (line 149). Discuss this.
Response: We have already discussed about this matter in section 3 (lines 419-450).
10) “comercial ulvan … a strange results” (line 187), please explain properly.
Response: We have already discussed about this matter in section 3 (lines 505-518).
11) Figure legends. Explained a, b and c fig but not the d fig. Legends in fig 3 and 4 identical.
Response:
11.1) Figure 3d and 4d were already mentioned in section 2.4.1.
11.2) Legends of Figure 3 and Figure 4 are not identical, since it slightly differs from each other by indication of “-WSSV solutions” and “-YHV solutions”; line 203 and 235, respectively.
12) Cannibalism is indicated as probably (l 244). How can this fact affect the results?
Response: Some dead shrimp in 1 and 5 mg/ml Ui HWEC-WSSV groups are caused by cannibalism, not from WSSV infection and moribund shrimp did not show clinical signs. Additionally, we always confirm these results by employing qRT-PCR/qPCR from moribund stages depending on pathogens tested.
13) Quantification of detected damages should be done by image analysis. Histopathology is of 2 groups and there are 4 in results (line 321).
Response: In this part, histopathological analyses were employed to indicate and confirm truly clinical signs of diseases caused by WSSV or YHV. Only some tested shrimp showing infected conditions or health status were selected to conduct this analysis. Accordingly, samples from these two groups showed specific alterations of each virus and no histopathological changes, respectively. Because of the limitation of space, we designed to show only the results of histology from the diseased shrimp.
14) Discussion: Why were selected these HWCEs concentrations or such incubations times?
Response:
14.1) Information from the preliminary test, we used the HWCEs concentrations ranged from 0.001, 0.01, 0.1, 1, 5 and 10 mg/ml against target viruses. We found that the concentrations of HWCEs at 1, 5 and 10 mg/ml exhibited the best survival rates and these concentrations are reasonable for application in shrimp culture in further studies.
14.2) For the incubation times, we adjust the incubation time according to the type of pathogens and the kind of HWCEs of green seaweed based on the methods of research from Balasubramanian et al. (2007) and Palanikumar et al. (2018) that we referenced.
15) Line 580 “ as described in 4.4.” and it was not described.
Response: We have already deleted this information.
Reviewer 3 Report
The subject of this research (bioactive compounds with antiviral properties from marine algae) is line with the themes of Marine Drugs and has a potential value for farmers by proposing an ecofriendly method to prevent disease outbreaks in aquaculture. Some minor revisions are, however, needed before its publication in Marine Drugs, as follows:
Title: change analyses into activities
line 18, Ulva armoricana
line 27, was further confirmed by the reduction in viral loads
line 44, rod- to elliptical- shaped
line 54, Furthermore, other main causes of diseases in shrimp culture systems is attributed to vibriosis, that recognize as aetiological agents 11 species of.....
line 62, delete To overcome these problems
line 76, Within marine macroalgae metabolites,
line 78, biological molecules contained in large amounts of marine extracts
line 87, pathogens and reduce the use of antibiotics
lines 106, 108, 110 avoid to repeat the numbers reported in table 1
line 126, had an infrared spectrum similar to those
line 145, No MICs displayed positive responses to extracts
line 146, Slight inhibition
line 155, the word preincubated solutions sounds strange, I would suggest extracts-virus mixture
line 178, the highest mortality rates of
line 180, similar to the Ui experiment, in the Cl experiment the injection of extracts from Cl.
line 185, groups with mortality rates of ...
line 186, injected group (Fig. 3d) exhibited
Captions to Figures 3 and 4, please include a description for letter 3d or 4d
Captions to Figures 3, 4: Cumulative mortality
The labels of x and y-axis of Fig. 5 and 6 are too small please improve them or convert these graphs into Tables.
Author Response
Reviewer 3
The subject of this research (bioactive compounds with antiviral properties from marine algae) is line with the themes of Marine Drugs and has a potential value for farmers by proposing an ecofriendly method to prevent disease outbreaks in aquaculture. Some minor revisions are, however, needed before its publication in Marine Drugs, as follows:
Response: Thank you for your comments.
1) Title: change analyses into activities
Response: We have already changed the title to “Antibacterial and antiviral activities of local Thai green macroalgae crude extracts in Pacific white shrimp (Litopenaeus vannamei)”.
2) line 18, Ulva armoricana.
Response: We already edited on line 18 and elsewhere.
3) line 27, was further confirmed by the reduction in viral loads
Response: We already edited that sentence (lines 27-28).
4) line 44, rod- to elliptical- shaped
Response: We already edited (line 44).
5) line 54, Furthermore, other main causes of diseases in shrimp culture systems is attributed to vibriosis, that recognize as aetiological agents 11 species of.....
Response: We already edited that sentence (line 56-57).
6) line 62, delete “To overcome these problems”.
Response: We already deleted (line 64).
7) line 76, Within marine macroalgae metabolites,
Response: We already edited (line 78).
8) line 78, biological molecules contained in large amounts of marine extracts
Response: We already edited (line 80).
9) line 87, pathogens and reduce the use of antibiotics
Response: We already edited (line 89).
10) lines 106, 108, 110 avoid to repeat the numbers reported in table 1
Response: We have modified this part to minimize the repetition of the number in Table 1 (line 111-112).
11) line 126, had an infrared spectrum similar to those
Response: We already edited (line 129).
12) line 145, No MICs displayed positive responses to extracts
Response: We already edited (line 148).
13) line 146, Slight inhibition
Response: We already edited (line 149-150).
14) line 155, the word preincubated solutions sounds strange, I would suggest extracts-virus mixture
Response: We well agreed to change to “extracts-virus mixture” (line 159).
15) line 178, the highest mortality rates of
Response: We already edited (line 180).
16) line 180, similar to the Ui experiment, in the Cl experiment the injection of extracts from Cl.
Response: We already edited that sentence (line 184).
17) line 185, groups with mortality rates of ...
Response: We already edited (line 188-189).
18) line 186, injected group (Fig. 3d) exhibited
Response: We already edited (line 190).
19) Captions to Figures 3 and 4, please include a description for letter 3d or 4d
Response: We already added the description of figure 3d and 4d on lines 202 and 234, respectively.
20) Captions to Figures 3, 4: Cumulative mortality
Response: We already changed this information on line 201 and 233, respectively.
21) The labels of x and y-axis of Fig. 5 and 6 are too small please improve them or convert these graphs into Tables.
Response: We already improved their graph sizes in Figures 5 and 6.
Round 2
Reviewer 1 Report
Manuscript arrangement; I do propose that materials and methods should come right after the introduction. Having the results after the introduction leaves one wondering how the work was done.
L574; free from should replace which freed from
L575; insert the word collected from
Author Response
Reviewer 1
Manuscript arrangement; I do propose that materials and methods should come right after the introduction. Having the results after the introduction leaves one wondering how the work was done.
Response: Thank you for your comments. We were unable to rearrange our manuscripts as your suggestion because Marine Drugs has a specific format that we have to arrange the heading topics accordingly.
L574; free from should replace which freed from
Response: We already edited your suggestion (yellow highlight).
L575; insert the word collected from
Response: We already edited your suggestion (yellow highlight).
Reviewer 2 Report
The authors have improved the manuscript that now coudl be accepted for publication.
Author Response
Reviewer 2
The authors have improved the manuscript that now could be accepted for publication.
Response: Thank you for your comments.